# Perceptions of male partners on maternal near-miss events experienced by their female partners in Rwanda

Patrick Gatsinzi Bagambe[1]*, Laetitia Nyirazinyoye[2], David Floyd Cechetto[3], Isaac Luginaah[4]

1 Department of Obstetrics and Gynecology, School of Medicine and Pharmacy, University of Rwanda, Kigali, Rwanda, 2 School of Public Health, University of Rwanda, Kigali, Rwanda, 3 Department of Anatomy and Cell Biology, University of Western Ontario, London, Ontario, Canada, 4 Department of Geography, Western University, London, Ontario, Canada

* patrickgatsinzi.pg@gmail.com

**Data Availability Statement:** All relevant data are within the paper and its Supporting Information files. If further access is needed, data are available from Prof. David F. Cechetto, the Director of the

## Abstract

### Background

Maternal near-miss refers to women who survive death from life-threatening obstetric complications and has various social, financial, physical, and psychological impacts on families.

### Objective

To explore male partners' perceptions of maternal near-miss experienced by their female partners and the associated psychosocial impacts on their families in Rwanda.

### Methods

This was a qualitative study involving 27 semi-structured in-depth interviews with male partners whose spouses experienced a maternal near-miss event. Data were analyzed using a thematic coding to generate themes from participants' responses.

### Results

Six key themes that emerged were: male partner's support during wife's pregnancy and during maternal near-miss hospitalization, getting the initial information about the spouse's near-miss event, psychosocial impacts of spouse's near-miss, socio-economic impact of spouse's near-miss, post- maternal near-miss family dynamics, and perceived strategies to minimize the impacts of near-miss. Male partners reported emotional, social, and economic impacts as a result of their traumatic experiences.

### Conclusions

The impact of maternal near-miss among families in Rwanda remains an area that needs healthcare attention. The residual emotional, financial, and social consequences not only affect females, but also their male partners and their relatives. Male partners should be involved and be well-informed about their partners' conditions and the expected long-term

"Training Support Access Model" (TSAM) project in Rwanda and Burundi, via the email: cechetto@uwo.ca.

**Funding:** This study was funded exclusively by Global Affairs Canada with cash and in-kind contributions from Western University, York University, University of British Columbia and Dalhousie University through the Training Support and Access Model (TSAM) for maternal, newborn and child health in Rwanda. The funder role in this study was only limited to financial support. The funders had no role in study design, data collection and analysis, decision to publish, or preparation of the manuscript.

**Competing interests:** The authors have declared that no competing interests exist.

effects of near-miss. Also, medical and psychological follow-up for both spouses is necessary for the enhancement of the health and well-being of affected households.

## Background

Globally, maternal mortality has remained a significant issue in many developing countries, where about 99% of all maternal deaths still occur (World Health Organization, 2019) [1]. Women in sub-Saharan Africa (SSA) in particular, are at heightened risk of maternal morbidity and mortality because of the inherent geographic disparities in the access and utilization of health care services [2, 3]. Hence, maternal mortality remains high across the continent, especially in countries where the health systems are still having multiple challenges. Reducing maternal and neonatal mortality remains a big issue in SSA as most deaths tend to be caused by obstetric complications that can be prevented with the effective utilization of health care services including the timely use of antenatal care services with skilled attendance throughout the continuum of pregnancy, childbirth, and the postnatal period [1, 4]. In Rwanda, despite enormous achievements in the millennium development goals, the maternal mortality ratio is still high and estimated to be 203 per 100,000 live births in 2020 with a slight decline from 210 per 100,000 live birth in 2015 [5, 6]. The lack of access, utilization of health care, and the persistence of home deliveries by some women has been suggested to be contributing to high number of maternal mortality due to complications during labor and/or birth. For instance, various studies done at the tertiary level hospitals showed that the incidence of the near miss was about 11 per 1000 live birth [7–10].

Maternal near-miss is described as "a woman who nearly died but survived a complication that occurred during pregnancy, childbirth or within 42 days postpartum or post termination of pregnancy" [11]. Maternal near-miss occurs when a woman survives death from life-threatening obstetric complications and this is frequently associated with obstetric complications like obstetric hemorrhage, unsafe abortions, eclampsia, obstructed labor, and sepsis [12, 13].

Maternal near-miss has various psychosocial, financial, and physical impacts on the affected families. Although the findings on the effects of male partners' presence during childbirth remain equivocal, some studies have reported that the involvement of male partners during childbirth contributes to a positive maternal outcomes such as emotional stability, lower levels of postpartum depression, more positive labor experience, and faster recovery through the enhanced adherence to medical care and provision of nutritional support to the woman immediately following childbirth [12–14]. Men who are culturally expected to be emotionally strong to support children and wives during and after the near-miss event, the high intensity of anxiety can last longer and interfere with daily activities and quality of life [15, 16]. The costs imposed by near-miss events are generally higher when compared to the cost of routine health care because, in many cases, the correction of the morbidity requires more specialized multidisciplinary care and mobilization of advanced equipment such as an anesthesia machine [17]. Even when healthcare cost is covered, near-miss complications can still lead to a vicious cycle whereby morbidity, loss of productivity, and poverty exacerbate each other [18]. In many contexts, delays in accessing and utilizing health care can result in morbidities and disabilities that further reduce economic productivity thereby acting as predisposing factors or catalysts of ill health, poverty, and destitution [2, 3, 7–9, 18].

In Rwanda, various quantitative studies have been conducted in different health settings to investigate aspects of maternal mortality and morbidity related to maternal near-miss complications and outcomes [19, 20]. In a previous qualitative study on women's perception on near-miss events, women have acknowledged the support of their male partners emotionally, and

financially, and by exempting women's domestic task to enhance a smooth recovery from the near-miss event [14]. What has not been done is an exploration of the perceptions of male partners of mothers who experienced the maternal near-miss on the social impact to the family and how can be used to develop strategies for health policy. In response, this study aimed at exploring the perceptions of males on maternal near-miss events experienced by their female partners. Specifically, the study will: explore males understanding of maternal near-miss events experienced by their spouses; assess the participation and role of male partners in preventing maternal near-miss events; explore the potential psychosocial and socioeconomic effects of the maternal near-miss events on families and explore participants' perceptions of what can be done to improve the impacts of maternal near-miss on Rwandan families.

## Methodology

This was a qualitative study that used a phenomenological approach to explore the reception of males on the maternal near-miss event in the Northern and Southern provinces of Rwanda. This research is part of a larger research and training program known as the Training, Support & Access Model (TSAM) for Maternal, Neonatal, and Child Health (MNCH) in Rwanda. The focus of TSAM project was to improve Maternal, Newborn, and Child Health (MNCH) in Rwanda through mentorship. TSAM is a collaborative partnership between the University of Western Ontario, the University of Rwanda, the Rwanda Medical and Dental Council, and the National Council of Nurses & Midwives in Rwanda. Other collaborating organizations include the Rwanda Society of Obstetricians & Gynecologists, the Rwanda Pediatric Association, the Rwanda Association of Midwives, and the Rwanda Society of Anesthesiologists. The program builds on the goal of the Rwanda Ministry of Health to improve the safe delivery of emergency care in MNCH in Rwanda through mentorship.

The Rwandan health system is structured in a way that women with pregnancy complications are transferred to referral hospitals which are mostly in urban areas where they can benefit from gynecologist experts and holistic care including advanced life support by an anesthesiologist. This is because the health centers are not staffed by doctors and the district hospitals have only 0 to 2 gynecologists with no intensive care or high-dependence units. With the subvention from TSAM, the gynecologists, anesthesiologists, and pediatricians are mobilized for a temporary attachment at lower district hospitals where they offer on-job mentorship and manage women with acute pregnancy complications that would otherwise lead to death (near-miss) in the ambulance during the transfer process. The selection of Northern and Southern provinces was based on TSAM mentorship that only covers the Northern and Southern provinces and the study recruited male partners of women who had a near-miss event during the period from February 2019 to February 2020 and was managed by the TSAM team of specialists in district hospitals without requiring a transfer to the referral hospital. Data were collected from April to July 2020. We estimated a sample size of 29 participants based on the TSAM registry of near-miss events, and on a prior qualitative study among the female partners of these males [14]. However, during the recruitment, two men had left the country and one was out of reach. Hence, we recruited 27 men in the study. The saturation was achieved by 24 participants' interviews and we added 3 more with no new emerging themes, hence saturation was confirmed. The average time from near-miss event to the data collection was 9 months (ranging between 5 to 13 months).

Because there was no hospital record of male partners at the hospital, we tracked the males based on the women's diagnosis from the TSAM registry. Once a woman with near-miss was identified from the TSAM registry, we checked the hospital registry to identify her local village and we used the Community Health Worker's (the Agent de "Santé Maternelle" (ASM))

database to identify the contact number of the community health worker (CHW) responsible for maternal health in that village and then called her/him to confirm the existence of the individual in the village. The community health workers in Rwanda are empowered to know every pregnant woman in their village, remind them to make antenatal contacts, teach them about danger signs of pregnancy, know the pregnancy outcomes, and do a postpartum and postnatal follow-up to identify women who need further attention at a health institution, which alleviates the shortage of healthcare personnel [21–23]. In each village, the CHW-ASM led the team to the woman's household to arrange for an interview with the male partner. Most of the women and their male partners did not have cell phones, but all CHW-ASM had cell phones provided by the government for their work-related communication and hence this facilitated the reach out to the study participants. In the few circumstances where the woman had a phone contact registered with her identification at the hospital, it was used to call her and guide the research team to her village.

Given that the information required was sensitive covering emotional and personal themes, we opted for face-to-face in-depth interviews and an interpretive approach to evaluate the research objectives. The in-depth interviews were conducted in Kinyarwanda, the local language, and led by the lead researcher who is experienced and multilingual. An interview guide of topics related to male partners' perceptions on maternal near-miss, its social and economic impact on the family, and what they think was done to improve the situation for their families was used. Prior to data collection, the interview guide was pre-tested by the lead researchers for clarity, organizational flow, and length. The interview guide was designed to be flexible to allow the interviewer to follow up on issues raised by participants that were not on the guide. Care was taken to ensure that the confidentiality of sensitive personal details was maintained. With the consent of participants, audio records of interviews were taken in order to capture as more detailed information as possible and field notes were used as needed. The recorded audios were then transcribed verbatim in Kinyarwanda and later translated into English.

A reflexive thematic data analysis was used to analyze the data. A coding system (codebook) was developed based on the research objectives and codes that emerged from data analysis from the in-depth interviews to facilitate the interpretation of responses provided by the participants. The coding and analysis of data were done using NVivo software version 12. The initial themes were nine and they captured the patterning of meaning across the transcripts. They were later revised, reviewed, and adjusted to form six broad themes tied to the data. The themes, subthemes, and codes are presented in Table 1. The lead researcher and research assistant were the coders and, worked independently on the coding system and each of them developed their codes. These codes were shared with each other for comparison, and a list of codes was agreed upon and harmonized to ensure inter-rater reliability of the study. A total of 64 meaning units were extracted, aggregated, and condensed into codes. A list of 25 final codes was then generated, followed by the emerging six themes. The lead researcher and research assistant separately developed emergent themes, and these themes were also harmonized into one list of themes that were considered for analysis, with different data categories generated based on the study objectives and the findings from the in-depth interviews. Following Bringer et al, the researchers constantly reviewed the emerging data to cross-check the validity of the analysis and to ensure the voices of the study participants are well-captured in the emerging themes, and for interpreting and explaining the findings [24].

This study was presented to the Institutional Review Board of the College of Medicine and Health Sciences at the University of Rwanda and ethical approval obtained before data collection. All participants were asked to sign an informed consent form approving their participation in the study. Participation in the study was voluntary after a full explanation of the study's purpose and the need for voice recording.

**Table 1. Themes, subthemes, and codes.**

| Themes | subthemes | Codes |
|---|---|---|
| Male partner's support during wife's pregnancy and during maternal near-miss hospitalization | • Ensuring wife's pregnancy is not in danger<br>• Provision of Insurance | • Roles during pregnancy<br>• Marking sure there is a health insurance for the family<br>• Not carrying heavy loads, and ensuring that wife<br>• Ensuring presence of food stock |
| Getting the initial information about the spouse's near-miss event | • Relatives and CHW communicating to the husband<br>• Personal witnessing of the developing near-miss event<br>• HCP giving information to the husband | • The husband was not home at the event<br>• The husband had cellphone but no the mother<br>• The mother called the CHW and the CHW informed the husband<br>• A relative or neighbor was called<br>• The couple was well aware that the pregnancy was high-risk<br>• Appreciation of the communication with HCP |
| The psychological impact of spouse's near-miss | • Feeling guilty for making the spouse pregnant<br>• Emotional dilemma for the situation and the coming future<br>• Discover of the importance of neighbors | • Emotional distress<br>• Feeling guilty<br>• Unclear future in case of spouse death<br>• Uncertainty about future pregnancies<br>• The sympathy of neighbors |
| Socio-economic impact of spouse's near-miss | • The family's source of income was vended<br>• Burden to the husband and the children | • Family animals and parcels were sold to cover the hospital bill<br>• The husband stopped the job to care for the wife<br>• Pregnancy was unintended<br>• Children tasked to care for the younger ones and miss schools |
| Post-maternal near-miss family dynamics | • Families struggling to recuperate the lost properties | • To buy new domestic animals<br>• Restore normal school attendance for the children<br>• Enhance recovery of the spouse |
| Perceived strategies to minimize the impacts of near-miss | • Lesson learned<br>• Advice for other men and health managers | • Create more human resource<br>• Aways secure a health insurance for the family<br>• Utilize family planning<br>• Support the wife to utilize maternal health services |

## Results

In this study, twenty-seven males were interviewed and they all had a female partner with a history of maternal near-miss. The interviews took an average of 45 minutes each. The respondents were mostly peasant farmers (25 participants) in their middle adulthood before age of 40 (20 participants). By the time of data collection, one male was already widowed but the female partner died from another cause many months after she had survived the maternal near-miss event. More details on the sample characteristics are summarized in Table 2.

The near-miss event exhibited by the female partners of males in this study were complications following pregnancy conditions such as ectopic pregnancy, placenta previa, abortion, vaginal delivery, uterine rupture, or cesarean delivery at term. All women survived the event but only nine neonates survived out of twenty neonates who were born at the gestational age of viability. A combination of criteria for the near-miss event were present in most women but the need for transfusion was the most prevalent among 25 women. The management had included life-saving hysterectomy and repair of bladder injury.

Currently, there is no data on the percentage of men attending childbirth in Rwanda. In many cases, the male partners are present at the health facility during childbirth, but they are not allowed into the labor or delivery room mostly because the room is shared by many women, with each of them needing privacy. Among the four males whose female partners had labor, three of them attended the labor but only one attended both the labor and the delivery.

**Table 2. Characteristics of the sample population.**

| Variable | | Frequency |
|---|---|---:|
| Age | 25–29 | 5 |
| | 30–34 | 7 |
| | 35–39 | 8 |
| | >39 | 7 |
| Occupation | Peasant farmers | 25 |
| | Mining | 1 |
| | Pastor | 1 |
| Marital status | Single | 2 |
| | Married | 21 |
| | Cohabitate | 3 |
| | Widowed | 1 |

It is important to note that the context of attending labor in Rwanda does not necessarily mean that the husband attended the labor room but that he was in the maternity unit during the time when the woman was in labor. Eighteen participants reported that when the near-miss event occurred, they were present at the hospital with their female partners, while others were not at the hospital. Eight of the nine participants reported that they had left the hospital either to care for the children left at home or to do other domestic work. One male reported that by the time his wife had a near-miss event, he was in another distant hospital being treated for a bone fracture resulting from a road traffic accident. The time interval from maternal discharge to the interview with the male partner was 12 to 24 months among 12 participants but some were interviewed after two years. More details on the characteristic of the near-miss events are provided in Table 3.

The six themes that emerged from the thematic analysis were: (i) male partner's support during wife's pregnancy and during maternal near-miss hospitalization, (ii) getting the initial information about the spouse's near-miss event, (iii) psychological-social impacts of spouse's near-miss, (iv) socio-economic impact of spouse's near-miss, (v) post-maternal near-miss family dynamics, and (vi) Perceived strategies to minimize the impacts of near-miss.

## Male partner's support during wife's pregnancy and during maternal near-miss hospitalization

Some of the male partners reported that they have been supporting their wives during pregnancy by playing important roles such as reminding them and taking them to antenatal care services in order to improve the pregnancy experience for the pregnant wives. Some of the key roles that were reported included making sure there is a health insurance for the family, that their pregnant wife was not carrying heavy loads, and ensuring that wife had good nutrition. In the comments below, participants talked about their involvement with their wives' pregnancies:

*When she was pregnant, I prevented her from doing heavy work, I kept finding the necessary diet for her, I arranged the health center appointments and I also helped her to implement the instructions from the health workers. [#114]*

*They had given us the expected date of delivery and, one day before, I reminded her to go to the health center. She refused and said she will rather wait for her contractions for so she can go to the health center. I said "no" and I insisted and I even accompanied her to the health*

**Table 3. Characteristics of the near-miss event.**

| Responses | | Number of mentions |
|---|---|---:|
| Index pregnancy¶ | Ectopic | 3 |
| | Placenta Previa | 2 |
| | Vaginal delivery | 4 |
| | Abortion | 2 |
| | Uterine rupture | 9 |
| | Caesarean Delivery | 7 |
| Neonatal survival | Yes | 9 |
| | No | 11* |
| | Not Applicable | 7 |
| Type of near-miss event¶ | Uterine rupture | 9 |
| | Hemorrhage | 25 |
| | Peritonitis& sepsis | 1 |
| | Eclampsia | 2 |
| Management offered¶ | Hysterectomy | 7 |
| | Transfusion | 25 |
| | Bladder injury repair | 3 |
| | Uterine repair | 14 |
| | Salpingectomy | 3 |
| The male attended labor | Yes | 3 |
| | No | 1 |
| | Not Applicable | 23 |
| The male attended delivery (vaginal or caesarean) | Yes | 1 |
| | No | 12 |
| | Not Applicable | 14 |
| Was at hospital at time of MNM diagnosis | Yes | 18 |
| | No | 9 |
| Interval from maternal discharge to interview | 0–6 months | 2 |
| | 6–12 months | 4 |
| | 12–24 months | 12 |
| | 24–36 months | 9 |

MNM: Maternal near-miss. *Includes 2 babies that died at re-hospitalization, ¶data from TSAM registry. Not applicable means in the context where the index pregnancy ended at gestational age (below 20 weeks) or context (such as uterine rupture, caesarean delivery) that is not applicable for neonatal survival or labor respectively.

*center where they found that she was having high blood pressure and low amniotic fluid. They called an ambulance and we went to the hospital. I am lucky I insisted that she should go to the hospital. [#115]*

Overall, participants in this study reported that they played key roles in sustaining and supporting their female partners during the time of the near-miss hospitalization and after they were discharged from the hospital. In most cases, the male partners had to stop their daily activities to care for their children at home and their wives in the hospital. While caring for the children at home, some male partners also carried food and other essentials for their wife and her caretaker at the hospital. In the quote below one respondent said:

*All activities were suspended. I could not do anything because our children were too young and I had to care for them alone and for their mother who was in the hospital. I had to be carrying food to her every day. [#101]*

The male partners also expressed how they assumed responsibility in cases of unfortunate circumstances where a fetal life was lost, needing to be buried while their wife was still in the hospital under critical medical care. In the comment below, a participant described how they arranged a burial event without the grieving mother who was still in the hospital.

*Because the fetus had died and the mother was still in critical condition, I had to stop all my activities and go home for the burial of that demised newborn. [#110]*

In the context of potential negative outcomes of a spouse's pregnancy, participants discussed the vital role of communication and access to skilled health care providers by their wives. Overall, most participants talked about the changes have had to make in order to prevent near-miss events and maternal death of their wives during pregnancy. In the narrative below, a husband explains what he has done to support and facilitate easy communication for his wife during her current pregnancy.

*I always tell her not to do heavy physical activities. I bought her a cell phone so that, whenever she has any problem, she should first call our community health worker even before she informs me. I also accompany her on antenatal care visits in order to capture all the instructions given to her. [#117]*

Although the majority of male partners accompanied their women to the health facility, none of them stayed in the hospital as the main caretaker for their wives. They frequently had to rely on the healthcare providers, a wife's friend, a neighbor from the community, or a member of the husband's in-law family to stay in the hospital with their sick wife.

## Getting initial information about the spouse's near-miss

Male partners of women who had a maternal near-miss event were generally aware that their wives had developed serious pregnancy complications that could lead to death. In many cases, male partners were not at home when their spouses were experiencing complications and therefore were mostly informed through phone calls either before or after the woman was already in a health facility. Most women had to call the community health worker (CHW) when they first realized abnormal signs such as bleeding. In cases where the CHW was not available, the women called upon their relatives or neighbors to come and accompany them to the health facility. One male participant explained how he learned about the worsening condition of his wife:

*My wife called on phone that she was not feeling well. After some minutes, I was called by the community health worker that they were taking her to the hospital. [#105]*

Another participant whose family lives very far away from a hospital explained how his mother informed him about the severe condition of his wife who was already in a hospital as a precaution in order to avoid possible pregnancy complications given their families' distance to the nearest hospital:

*I was informed by my mom who was her next of kin. In fact, my wife had been in the hospital waiting for her due date because she had had multiple cesarean deliveries and the CHW warned us that even a simple contraction is a danger sign to her. So, we preferred to keep her in the hospital because we live far from the road. During that time, she was in hospital, she developed complications and that's when my mother called me. [#106]*

In the local communities, CHW-ASMs try to educate women on pregnancy issues and especially the signs of pregnancy complications women always have to keep in mind. Hence, some of the participants in this reported they were present when their wives reported to them that they are experiencing signs of complications as described to them by their CHW-ASM. Some participants reported they could clearly see their spouse was in severe pain and complications. One participant mentioned that:

*When it started, I could see it myself because she started with bleeding and she was in pain. . .and then I called the community health workers who took her to the health center. [#103]*

*When she left home, I could see myself that she was severely sick because she was already bleeding too much that we needed support from other people (porters) to take her to the hospital. [#102]*

Although most of the male partners in this study were quite aware when their wives were going through their near-miss complications, some participants indicated that although they knew their wives were admitted to the hospital, they had little information about their wives' condition could be life-threatening. In the comment below, a participant indicated that he felt his wife was in the hospital like other women:

*During that time, my wife was in hospital like other women. I did not know that she had some particularity but I only remember that we spent more days than other women. I am just learning from you and I thank you so much. Imagine if I had lost my wife. . .. [#121]*

Another participant explained that he only learned the severity of the wife's condition on the day of discharge.

*I did not know she ever had any complications until she was discharged from the hospital. I only knew that when someone is pregnant, anything may happen. [#123]*

It emerged from the findings that even though many husbands were informed about the life-threatening nature of their female partners' ill-health, some participants reported they were never informed about potential life intraoperative interventions the wives had to go through in the hospital. In the quote below, a participant reported that he was never told his wife was to undergo a life-saving hysterectomy, and neither was he told even after the operation.

*I don't know if it (the uterus) was removed or if she underwent a permanent sterilization but what I know is that she does not menstruate anymore and she is not getting pregnant [#102]*

### Psychological impact of spouse's near-miss

Overall, the participants talked about the psychological impact of their wives' near-miss events on them and their families. Most talked about the emotions and stress they went through

when their wives were at the hospital with complications of pregnancy and for some even after the wife was discharged. In many cases, male partners worried that they could lose their wives and/or their child. The emotions ran when a pregnant woman had to undergo more than one surgery while in hospital. A participant described how he was worried about whether his wife will recover but kept calling upon God for a miracle to save their wives' lives:

> *I was worried very much because she had been operated upon twice in one day and I thought it would cause her some permanent disability or she may even die. All I knew is that if some-one gets operated on two times in one day, she/he does not recover. But I kept praying to God to show his miracle again. It was very stressful [#108]*

Some participants reported that although they were not very worried during the hospitalization of their wives, most of them were worried of the challenges during the postpartum period when their wives are back home after their near-miss event. To prevent aggravating their wives already fragile conditions, some participants did not allow their wives "to perform regular activities such as carrying heavy load as this may rupture at the surgical site."

For some participants, the emotional distress also resulted in a sense of self-blame as some men expressed that they felt responsible for their spouses' health and apparent near-death situation. As indicated in the comment below, some also reported guilt for not engaging with their wives much earlier in the pregnancy to plan for eventualities.

> *I was feeling very much worried about the problems that she was having. I felt guilty that I am the cause of her problems given I was the father. I felt my future was going to be difficult as well. [#107]*

Furthermore, the near-miss experience whereby their wives almost lost their lives, also led some of the men to make a strong resolve for permanent contraceptive use. In the narrative below, a male partner explained their regret of having made their wife pregnant and wanting to experience a near-miss situation again:

> *I was affected psychologically. I feel like I don't want to make any child anymore unless the sterilization fails to work. I cannot afford the consequences that I have seen, and I don't want to risk the life of my wife. [#119]*

The psychological consequences of the near-miss event did not only affect the male partners but, also, extended to other relatives, particularly to those who assisted as caregivers during hospitalization because of pregnancy complications.

During the time of data collection, some couples had already delivered a healthy baby since their near-miss experience and were therefore relieved as the normal delivery was reassuring for them to pursue their normal lives and daily activities like before the near-miss event:

> *it has affected me psychologically. But I was eventually relieved because after that my wife got pregnant again and we have a new baby. I was worried that this baby would also die like it happened in the previous pregnancy. [#103]*

Since their near-miss experience, some of the couples that were pregnant at the time of the interviews talked about the actions they had taken to minimize any risk that could result in pregnancy complications. A participant talked about how he had taken the responsibility of domestic work to relieve his wife of any strenuous activity that could put her at risk:

*Yes, of course, it has affected me. She is now pregnant again and I keep stressing her every day not to forget any of her appointments at the health center. I do the household work and I don't allow her to lift heavy loads. I feel a little anxious that she may develop some other complications like in the previous pregnancy. [#117]*

## Socio-economic impact of spouse's near-miss

All the participants, except one, in this study reported the economic impact of the near-miss on their families. In most cases, the families had to sell their assets, notably cows, goats, and other domestic animals, or even land to afford the medical bills and for the post-operative recovery of the mother, and for the upkeep of the children at home. The majority of husbands reported having to stop their daily work while their wives were in the hospital:

*We only do subsistent peasant farming and at that time we suspended everything. We had even to sell our cows and we were never able to regain it after she was discharged from the hospital. This brought us economic hardships because we sold everything [#107]*

Another participant commented that:

*Our household income decreased, and I had to sell our land that had coffee plantation so that we can afford the medical bill. [#110]*

Other participants talked about how the impact of the maternal near-miss has compounded their household poverty and made them poorer. One participant recalled how they had to borrow money from their neighbors to be able to afford the medical bills and ever since, for almost two years, they have not yet been able to pay back what they borrowed:

*We were poor before but this time we became poorer. We borrowed money from our friends and we have not been able to pay all of it, but they understand. [#119]*

The preceding comment alludes to how participants in this study reported how some of their friends and neighbors were supportive during their near-miss predicament. Many participants reported that while their wives were at the hospital, and they had to be going to the hospital frequently, their relatives and surrounding neighbors took responsibility and were taking care of the young children and also helping them with some the domestic work such as feeding the animals. In the comments below, participants talked about the social support from their neighbors and expressed a sense of belonging in their local community given the kindness of their neighbors, and the fact they are ready to show them similar kindness should the need arise:

*Our friendship with my surrounding neighbors got better. They supported us and I also know that I also owe them a lot in case they are in trouble. [#102]*

*The situation that happened to my wife made me change the way I consider my neighborhood. They showed me love and I appreciate them for that. [#105]*

*The husband of my wife's sister and my parents were very helpful. They helped in taking care of her during hospitalization and they also helped in looking after our family. One of our children move to my parents' and another went to her maternal aunt. [#107]*

## Post- maternal near-miss family dynamics

During interviews with husbands whose wives had had a maternal near-miss, they shared the current situation in their households after their wife's recovery from the life-threatening event. By the time of this interview, most of the men indicated that their households had not returned to their normal routines mainly because the couple was still dealing with the consequences of the near-miss event. Because many of the women who experienced the near-miss complications were not yet able to work effectively, some of the participants reported having to take on extra work in order to sustain their families. In the comment below, a participant talked about the challenges he was going through to compensate for their wives' inability to help with the domestic work:

> *My wife cannot yet work as she used to. I am working much harder than before, trying to see if we can buy a cow again and also sustain my family as we used to do before that pregnancy. [#125]*

Another participant indicated how they appreciate the determination and resilience of their wife and how this has helped to strengthen their respect and love for her:

> *Now I work much harder for the family and for her because she is weak. This situation made me realize that my wife is probably stronger than me. . . because, if I were to bleed like she bled, I would have died. Currently, I respect her a lot more and I will not allow anyone to look down on her. [#116]*

For those whose wives recovered fast enough, they reported resuming their routine household activities with their wives. However, most of them agreed that on subsequent pregnancies, they will always insist on their wives to eat nutritious food, attend all the required antenatal visits, make sure they deliver with a skilled birth attendant and accompany them to the health facility each time.

## Perceived strategies to minimize the impacts of near-miss

Given the lessons learned from the consequences of maternal near-miss complications, some of the participants talked at length about the actions they have taken within their families to prevent future problems. Most of the participants reported using at least one method of modern family planning. Furthermore, all the participants also indicated they have now realized the importance of health insurance and have been advising their friends to try and get health insurance for their families. Together with health insurance, the participants also reported now identifying the need to always save some money as their responsibility to their family, since such money can be used for pregnancy and delivery expenses. Moreover, in the comments below, some of the men in this study warned young people not to rush into marriage unless they are well financially stable, and also for households to ensure they have health insurance:

> *I would advise the healthcare providers to work with the government so that they educate young people not to rush into marriage or get pregnant when they don't have enough money. [#112]*

> *My advice to other men is to have health insurance and to save money because you're never sure of when labor may start. [#120]*

The participants in this study also discussed what they thought were other issues that may be leading to the high rates of near-miss cases in their communities. The reasons that were mentioned included: the lack of full-time specialists at the district hospitals, the lack of readily available ambulance service to transport women, especially from remote areas to health centers, women not complying with health education given during antenatal contacts, and the lack of health equipment such as ultrasound machines at the health centers where the majority of women in rural areas seek antenatal care.

## Discussion

The aim of this study was to assess the perceptions of male partners whose spouses experienced maternal near-miss events and the impact maternal near-miss has on their families. The findings show that the immense trauma that was faced by women who experienced maternal near-miss was felt by their male partners in multidimensional ways including intense emotional distress and anxiety, health care access challenges, and social and financial impacts.

Overall, most of the men in this study agreed that support for their pregnant wives is something they have always done. As indicated in the findings, some males consistently reminded their spouses of the need to attend antenatal care services. In the unfortunate situations where their spouses experienced a near-miss event, the participants got information from varied sources, but mostly from their local CHW-ASM. At the onset of a near-miss event, male partners would call upon their ASM or call an ambulance. The ASM together with the support of the husband, if available, would help to take the woman to a health facility as soon as they can to prevent a possible irreversible life-threatening event at home. In Rwanda, CHW-ASMs have been involved in the provision of first-line primary health care services that, in most cases, have resulted in saving the lives of maternal near-miss patients, as they tend to assist women to the health centers and hospitals on time [8]. In this study, male partners played a role in supporting their spouses when they were hospitalized by either by being caretakers, delivering food to the hospital, caring for children at home, or working tirelessly to find the means during and after the maternal near-miss event. Redshaw and Henderson reported similar findings on fathers' engagement in pregnancy and childbirth in England [25].

Emerging out of this research, most participants were appreciative of the care and services that were given to their partners and attributed the survival of the partners from near-miss to the efforts of the healthcare staff. Nevertheless, consistent with earlier studies, most of the participants also reported that they did not get adequate information regarding their spouses' conditions and prognosis when they were admitted to the hospital with a maternal near-miss condition [12, 26–28]. For instance, some participants highlighted that during hospitalization, the communication between healthcare workers and the patients' families was lacking, hence the husbands were never engaged in decision-making for their wives who were in critical condition and this reinforced their own anxiety. These negative experiences could be avoided if healthcare workers were providing appropriate communication to the husband about the status of their wife and whatever issues they may need to know. This finding is in agreement with the results from other studies which indicated that most of the time, male partners feel excluded by healthcare providers in the decisions during childbirth and these partners consider this as unfair [29–33]. To the extent that certain major procedures (e.g., hysterectomy) were performed on some near-miss patients without their spouse's knowledge and this further worsened the emotional distress of the male spouses. In previous TSAM studies, the lack of communication between healthcare workers and patients was attributed to the fact that healthcare staff in this resource-limited setting may be overstretched in their duties [3, 14]. Despite the existing challenges, healthcare professionals need to be communicating with patients and

their caretakers about their health conditions, interventions to be done, and what they should expect in the future due to the developed health condition.

While healthcare providers may be busy trying to save the lives of women experiencing near-miss, the lack of engagement with male partners may precipitate negative consequences that result in mounting anxiety as reported by the participants. Hence, the psychological impact of maternal near-miss was frequently cited as a major outcome. Male partners described their experiences including emotional and traumatic stress that continued to interfere with their everyday lives even after the event. Consistent with previous studies, several male partners expressed their fear of loss due to the maternal near- miss event experienced by their partners [12, 29, 34]. Unfortunately, the effects of psychological distress have the potential to affect the couple's relationship which can result in other marital problems [35].

The extended stay in hospital by spouses who experienced near-miss resulted in socioeconomic challenges to the families. Several of the participants talked about the resulting financial hardship after the near-miss, the impact of which led some families to sell their animals and land or to take loans in order to cover the hospital bills and the cost of living in the post-near-miss discharge from the hospital. The unavailability of participants to attend their farms, especially during the cultivation season, also further compounded families' problems related to the lack of food in their households. These findings are consistent with previous studies documenting that, households that experience a maternal near-miss event, face a large financial burden due to the cost of obstetric emergencies while the families are not financially producing [10, 36].

The enduring importance of family and community support was also revealed in this study. Many of the participants reported how they benefited from the invaluable support of their neighbors in their communities. The financial and social support they received helped them to cope with the near-miss situation and the associated aftermath in their families. These findings are consistent with the work by Mbalinda et al in rural Uganda where male partners of women who experienced maternal near-miss expressed that coping with the trauma of their spouse's near-miss was moderated by the support they received from their friends, family members, and relatives [12].

An important finding in this work is the support that the husbands give to their wives following the maternal near-miss event. In order to avoid future pregnancy complications, some participants reported supporting and ensuring that their wives go for long-term contraception. Others also took it upon themselves to be more involved in their wives' antenatal visits following subsequent pregnancies. The increasing involvement of husbands in their wives' reproductive health processes has also been reported by other scholars in various countries [13, 37–40]. In the Rwandan context, this is a shift away from the traditional belief system that pregnancy-related matters are a "women's only" area.

Given the design of this interpretative study, there were some limitations. For instance, as a result of our study design, there is a likelihood of bias in the participant selection. We only asked to talk to families that had experienced maternal near-miss and hence, there was no comparative group of men in terms of their perceptions. Furthermore, the interviews with male partners took place several months after the near-miss event, hence, there could have been a recall bias by the participants.

## Conclusion

Despite the limitations of this study, the findings are important for improving maternal and child health care providers in Rwanda and elsewhere in sub-Saharan Africa. A good maternal health outcome from a life-threatening obstetric condition such as a near-miss can be

reassuring and a good indicator of the role of having specialists' mentors at district hospitals. This therefore calls for more residency training, and efforts to sustain the mentorship at district hospitals to for the prevention of prevent maternal deaths. The prevention of maternal deaths is not only beneficial to the mothers only but also to their male partners, offspring, families, and local communities at large. As indicated in this study, the overall consequences of a near-miss event can be emotionally, financially, and socially devastating. Consequently, male partners should always be included earlier in the decision-making process whenever it is possible and especially there should be clear communication guidelines whereby healthcare workers will be required to communicate with partners and caregivers on the health conditions and the expected long-term outcomes of near-miss patients. Furthermore, given the traumatic experience that is usually experienced not only by the patient but also by their spouse and other family members, the Rwandan healthcare system should aim at implementing a program that focuses on providing extensive and regular medical and psychological follow-up of maternal near-miss patients and their families. This can possibly be done using the CHW-ASM program if training programs are provided. Additionally, the deployment of more specialists to district hospitals will help to provide early interventions in case of complications. This can be reinforced by availability of ambulance services, especially in rural areas, together with equipment such as ultrasound and trained health personnel skilled in using them at the health center level for early recognition of high-risk pregnancies and for mitigation plans to prevent negative outcomes of maternal near-miss.

## Supporting information

**S1 File. Consent form for participation in the research study on Barrier challenges of utilization of maternal health services in Rwanda.**
(DOCX)

**S2 File. Interview guide form for participation in the research study on Barrier challenges of utilization of maternal health services in Rwanda.**
(DOCX)

## Acknowledgments

We acknowledge Dr Polyphile Ntihinyurwa for his valuable assistance in transcription, translation, and manuscript preparation during this study.

## Author Contributions

**Conceptualization:** Patrick Gatsinzi Bagambe.

**Data curation:** Patrick Gatsinzi Bagambe.

**Formal analysis:** Patrick Gatsinzi Bagambe.

**Funding acquisition:** Patrick Gatsinzi Bagambe, David Floyd Cechetto.

**Investigation:** Patrick Gatsinzi Bagambe.

**Methodology:** Patrick Gatsinzi Bagambe.

**Project administration:** Patrick Gatsinzi Bagambe.

**Resources:** Patrick Gatsinzi Bagambe.

**Supervision:** Laetitia Nyirazinyoye, David Floyd Cechetto, Isaac Luginaah.

**Validation:** Patrick Gatsinzi Bagambe, Laetitia Nyirazinyoye, David Floyd Cechetto.

**Visualization:** Patrick Gatsinzi Bagambe.

**Writing – original draft:** Patrick Gatsinzi Bagambe.

**Writing – review & editing:** Patrick Gatsinzi Bagambe, Laetitia Nyirazinyoye, David Floyd Cechetto, Isaac Luginaah.

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
