## [Decision Letter · Decision Letter 0]

20 Jul 2022

PONE-D-21-39209Perceptions of male partners whose spouses experienced maternal near-miss in RwandaPLOS ONE

Dear Dr. Bagambe,

Thank you for submitting your manuscript to PLOS ONE. After careful consideration, we feel that it has merit but does not fully meet PLOS ONE’s publication criteria as it currently stands. Therefore, we invite you to submit a revised version of the manuscript that addresses the points raised during the review process.

Please note that we have only been able to secure a single reviewer to assess your manuscript. We are issuing a decision on your manuscript at this point to prevent further delays in the evaluation of your manuscript. Please be aware that the editor who handles your revised manuscript might find it necessary to invite additional reviewers to assess this work once the revised manuscript is submitted. However, we will aim to proceed on the basis of this single review if possible.  Your manuscript has been assessed by an expert reviewer, whose comments are appended below. The reviewer has highlighted concerns about several aspects of the methodology, clarity of the study aims, and overall quality of the scientific writing. Please ensure you respond to each point carefully in your response to reviewers document, and modify your manuscript accordingly. Please submit your revised manuscript by Sep 02 2022 11:59PM. If you will need more time than this to complete your revisions, please reply to this message or contact the journal office at plosone@plos.org. Please include the following items when submitting your revised manuscript:A rebuttal letter that responds to each point raised by the academic editor and reviewer(s). You should upload this letter as a separate file labeled 'Response to Reviewers'.A marked-up copy of your manuscript that highlights changes made to the original version. You should upload this as a separate file labeled 'Revised Manuscript with Track Changes'.An unmarked version of your revised paper without tracked changes. You should upload this as a separate file labeled 'Manuscript'.

We look forward to receiving your revised manuscript.

Kind regards,

Joseph Donlan

Editorial Office

PLOS ONE

Journal Requirements:

Reviewers' comments:

Reviewer's Responses to Questions

**Comments to the Author**

1. Is the manuscript technically sound, and do the data support the conclusions?

Reviewer #1: Partly

2. Has the statistical analysis been performed appropriately and rigorously? 

Reviewer #1: N/A

3. Have the authors made all data underlying the findings in their manuscript fully available?

Reviewer #1: No

4. Is the manuscript presented in an intelligible fashion and written in standard English?

Reviewer #1: No

5. Review Comments to the Author

Reviewer #1: Overall, this qualitative study exploring male partners’ perspective on the impact of maternal near-miss events on the family has potential to be a welcome addition to the literature, however this is subject to meeting the necessary quality requirements.

1. In the background you state, “Different studies showed that the involvement of male partners during childbirth contributes to the positive outcomes and strengthen the relationship between male and female partners,(12,13).” Please clarify the type of outcomes referred to here – does this include near-miss events or does this concern other aspects? Literature concerning presence at childbirth is more mixed than indicated here.

2. You describe the study aim as concerning “the social impact on the family” (p2) however the study seems broader than this, for example also concerning psychological/emotional aspects. Please consider rephrasing the aim.

3. It would be helpful to provide some further detail about the study setting, for example you state that these two provinces were chosen because of relatively high prevalence without requiring transfer – please briefly provide the prevalence information for these provinces and the other provinces. It would be helpful for readers too to briefly have context around birthing in Rwanda eg what % birth at a facility, are these all public facilities. What percentage of fathers attend childbirth in Rwanda, if known? This is important contextual information. In places, the manuscript reads as though all of the fathers attended the birth whereas later text suggests that was not the case.

4. The methods do not provide adequate detail concerning the process of analysis. Please provide further information eg type of thematic analysis used; how the lead researcher and research assistant ‘harmonized’. This appears to be a descriptive analysis.

5. The methods do not provide adequate detail concerning recruitment. Amongst the households that the team were ‘led to’, for what proportion had the father attended the birth? How many fathers were asked to take part? How many declined? How long did interviews last (eg on average/range)? How recently had the near-miss event occurred? Concerning sampling, how was this done? What was the planned sample size? How was the final sample size determined?

6. Sample characteristics. None have been presented. Key information may include aspects such as whether or not they were at the birth, the type of near-miss event, whether the baby survived, parity.

7. Themes.

The theme concerning male partner support in pregnancy include an example about perinatal loss which does not appear to be part of this theme; the remainder of the theme appears (from the examples) to be about partners role in accessing/navigating healthcare (and possibly about ‘protection’). Perinatal loss is also raised in the later theme, concerning psychological impact.

There is unnecessary repetition in the theme on socio-economic impact and on family dynamics. For Consider relocating the content concerning socio-economic from the family dynamics section, to the socio-economic section.

8. Ethical considerations. One quotes indicates that the partner only learned about the near miss event through the process of the research. This needs to be discussed in the paper. Similarly, reflect on your comment early in the discussion that “most of the men in this study agreed that support for their pregnant wives is something they have always done”. It is unsurprising that there may be high levels of difficult emotions eg guilt following a near-miss event – as seen in literature following perinatal loss for example. May these fathers have felt ‘questioned’ about their involvement/support given the circumstances? It is unclear what is meant by the comment: “the participants were aware that the researchers were healthcare providers and this could have biased their responses in favour of their own expectations”.

9. Please clarify who you are referring to when stating in the abstract and conclusion: “Extensive and regular medical and psychological follow-up is necessary for the health and well-being of the affected households.” Does the medical follow-up only concern the mothers?

10. It does not seem appropriate to make the following statement (in the abstract of conclusion): “Good maternal outcomes from near-miss obstetric complications indicate a well-functioning healthcare system in Rwanda.”

11. There are typographical and grammatical errors that need to be corrected throughout which need to be addressed.

6. PLOS authors have the option to publish the peer review history of their article (what does this mean?). If published, this will include your full peer review and any attached files.

Reviewer #1: No

---

## [Author Response · Author response to Decision Letter 0]

10 Jan 2023

Patrick G Bagambe

Senior Lecturer of Obstetrics and Gynecology

University of Rwanda

patrickgatsinzi.pg@gmail.com

+250 788 302 804

August 23, 2022

Dear Sir/Madam

Re: PLOS-ONE-D-21-39209: “Perceptions of male partners on maternal near-miss events experienced by their female partners in Rwanda”. 

Please find enclosed a revised version of our manuscript number PLOS-ONE-D-21-39209. The Revisions are based comments and suggestions from yourself and the reviewers. The manuscript has been revised considerably. Please find enclosed a detailed response to the comments the manuscript reviewers.

The manuscript has not been submitted for consideration for publication elsewhere. There are no financial agreements or other conflicts of interest to disclose.

I look forward to your evaluation of the revised manuscript. 

Sincerely,

Patrick Bagambe

Response to Reviewers and editors' comments:

Title of the Paper:

In response to the reviewer’s comments we have revised the title of the paper from:

“Perceptions of male partners whose spouses experienced maternal near-miss in Rwanda” to

“Perceptions of male partners on maternal near-miss events experienced by their female partners in Rwanda”. 

Data availability Statement:

I accept to make the data available.

Specific Comments: 

Detailed responses to comments from the two manuscript reviewers are itemized in the enclosed package.

Comment 1. In the background you state, “Different studies showed that the involvement of male partners during childbirth contributes to the positive outcomes and strengthen the relationship between male and female partners (12,13).” Please clarify the type of outcomes referred to here – does this include near-miss events or does this concern other aspects? Literature concerning presence at childbirth is more mixed than indicated here.

Response: Thank you for this clarification. In response, we have revised this statement for clarity. The statement now reads: Although the findings on effects of male partner presence during childbirth remains equivocal, some studies have reported that the involvement of male partners during childbirth contributes positive maternal outcomes such as emotional stability, a relieved aftermath depression, realized birth expectations, and faster recovery through the enhanced adherence to medical care and provision of nutritional support to the woman immediately following childbirth.

2. You describe the study aim as concerning “the social impact on the family” (p2) however the study seems broader than this, for example also concerning psychological/emotional aspects. Please consider rephrasing the aim.

Response: We appreciate this suggestion. The aim of the study has been revised to read “This study aimed at exploring male partner’s perceptions of maternal near-miss experienced by their female partners and the associated psychosocial impacts on their families.”

3. It would be helpful to provide some further detail about the study setting, for example you state that these two provinces were chosen because of relatively high prevalence without requiring transfer – please briefly provide the prevalence information for these provinces and the other provinces. It would be helpful for readers too to briefly have context around birthing in Rwanda eg what % birth at a facility, are these all public facilities. What percentage of fathers attend childbirth in Rwanda, if known? This is important contextual information. In places, the manuscript reads as though all of the fathers attended the birth whereas later text suggests that was not the case.

Response: Thank you so much for this suggestion. In revised manuscript, we have clarified that the selection of Northern and Southern provinces was based on a project called “Training, Support & Access Model (TSAM) for Maternal, Neonatal, and Child Health (MNCH) in Rwanda”. The focus of the TSAM project was to improve Maternal, Newborn, and Child Health (MNCH) in Rwanda through mentorship and the project was funded by Global Affairs Canada. TSAM implemented a healthcare staff mentorship program that included the Northern and Southern provinces with goal of improving maternal health outcomes (Please see page 3 of the revised manuscript).

4. The methods do not provide adequate detail concerning the process of analysis. Please provide further information eg type of thematic analysis used; how the lead researcher and research assistant ‘harmonized’. This appears to be a descriptive analysis.

Response: We appreciate this suggestion. We have revised the methods section to provide more details on the analysis process (please see page 5 of the revised manuscript). A reflexive thematic data analysis was used to analyse the data. A coding system (codebook) was developed based on the research objectives and codes that emerged from data analysis from the in-depth interviews to facilitate the interpretation of responses provided by the participants. The lead researcher and research assistant worked independently on the coding system and each of them developed their codes. These codes were shared with each other for comparison, and a list of codes was agreed upon, harmonized to ensure inter-rater reliability of the study and a list of final codes generated, followed by the emerging themes. The coding and analysis of data were done using NVivo software version 12. The lead researcher and research assistant separately developed emergent themes, and these themes were also harmonized into one list of themes that were considered for analysis, with different data categories generated based on the study objectives and in-depth interviews findings. Following Bringer et al (2006), the researchers constantly reviewed the emerging data to cross check the validity of the analysis and to ensure the voices of the study participants are well-captured in the emerging themes, and for interpreting and explaining the findings. 

5. The methods do not provide adequate detail concerning recruitment. Amongst the households that the team were ‘led to’, for what proportion had the father attended the birth? How many fathers were asked to take part? How many declined? How long did interviews last (eg on average/range)? How recently had the near-miss event occurred? Concerning sampling, how was this done? What was the planned sample size? How was the final sample size determined?

 Response: Thank you very much for this clarification. We have revised the methods section of the manuscript to provide more detail and clarity regarding the study context, data collection and analysis. We also provided more detail on the recruitment process. We have added more detail on the number of fathers in the various sample categories. We also provided the average length on the interviews (45 minutes) and indicated that the time of near-miss event ranged from 5 to 13 months (Please see pages 3 to5 of the revised manuscript). 

6. Sample characteristics. None have been presented. Key information may include aspects such as whether or not they were at the birth, the type of near-miss event, whether the baby survived, parity.

Response: Thank you so much. We have included Tables 1 and 2 on sample characteristics accordingly. 

7. Themes. The theme concerning male partner support in pregnancy include an example about perinatal loss which does not appear to be part of this theme; the remainder of the theme appears (from the examples) to be about partners role in accessing/navigating healthcare (and possibly about ‘protection’). Perinatal loss is also raised in the later theme, concerning psychological impact. There is unnecessary repetition in the theme on socio-economic impact and on family dynamics. For Consider relocating the content concerning socio-economic from the family dynamics section, to the socio-economic section.

Response: We appreciate this suggestion. The information has been relocated into the appropriate section. Also, there were two themes on male partner’s support (one for during pregnancy and another for during maternal near-miss hospitalization) and were both merged into one entitled: Male partner’s support during wife’s pregnancy and during maternal near-miss hospitalization

8. Ethical considerations. One quotes indicates that the partner only learned about the near miss event through the process of the research. This needs to be discussed in the paper. Similarly, reflect on your comment early in the discussion that “most of the men in this study agreed that support for their pregnant wives is something they have always done”. It is unsurprising that there may be high levels of difficult emotions eg guilt following a near-miss event – as seen in literature following perinatal loss for example. May these fathers have felt ‘questioned’ about their involvement/support given the circumstances? It is unclear what is meant by the comment: “the participants were aware that the researchers were healthcare providers, and this could have biased their responses in favour of their own expectations”.

Response: Thank you for this comment. We have revised the manuscripts in response. We have clarified the importance of communication between health staff and patients/families in general with the Rwandan health system. The issue of poor communication between healthcare staff and male partners of women who experienced near-miss has been discussed and highlighted as a recommendation that needs to be addressed. We have clarified why male partners may have been feeling some guilt in relation to their partner’s near-miss outcome. The statement “the participants were aware that the researchers were healthcare providers” has been removed as it is redundant.

9. Please clarify who you are referring to when stating in the abstract and conclusion: “Extensive and regular medical and psychological follow-up is necessary for the health and well-being of the affected households.” Does the medical follow-up only concern the mothers?

Response: Thank you for this clarification. We have revised this statement to indicate the medical and psychological follow-up for both spouses is necessary for the enhancement of the health and well-being of affected households.

10. It does not seem appropriate to make the following statement (in the abstract of conclusion): “Good maternal outcomes from near-miss obstetric complications indicate a well-functioning healthcare system in Rwanda.”

Response: Thank so much for this suggestion. The statement has been removed.

11. There are typographical and grammatical errors that need to be corrected throughout which need to be addressed.

Response: We read the manuscript thoroughly and hope that we have corrected all the typographical errors. We are grateful for the time you took to read our paper.

---

## [Decision Letter · Decision Letter 1]

12 Mar 2023

PONE-D-21-39209R1Perceptions of male partners on maternal near-miss events experienced by their female partners in RwandaPLOS ONE

Dear Dr. Bagambe,

Thank you for submitting your manuscript to PLOS ONE. After careful consideration, we feel that it has merit but does not fully meet PLOS ONE’s publication criteria as it currently stands. Therefore, we invite you to submit a revised version of the manuscript that addresses the points raised during the review process.

Specifically these areas are in need of revision, clarification, and improvement:

The interview guide should be described in the methods section briefly.

A description of the coding tree with theme(s) should be provided.

The authors should state how many meaning units and codes were extracted from the data.

We look forward to receiving your revised manuscript.

Kind regards,

Forough Mortazavi

Academic Editor

PLOS ONE

Journal Requirements:

Additional Editor Comments (if provided):

Thank you for your revisions. This manuscript is dealing with an important subject. Therefore, to improve it and receive deserved attention from the readers, some revisions are needed.

In page 4, the authors state, ‘Once a woman with near-miss was identified from the TSAM registry, we checked the hospital registry to identify her local village and we used the Community Health Worker’s …….’ But in the limitations of the study we find that the interviews were conducted months after discharging the women from the hospital. For the sake of clarity, the interval between the discharge of a mother from hospital and the interview with her spouse should be included in table 1 or 2. As men’s demographic data is included in the text, PLS remove it from the supplementary files.

Reviewers' comments:

Reviewer's Responses to Questions

**Comments to the Author**

1. If the authors have adequately addressed your comments raised in a previous round of review and you feel that this manuscript is now acceptable for publication, you may indicate that here to bypass the “Comments to the Author” section, enter your conflict of interest statement in the “Confidential to Editor” section, and submit your "Accept" recommendation.

Reviewer #1: (No Response)

2. Is the manuscript technically sound, and do the data support the conclusions?

Reviewer #1: Yes

3. Has the statistical analysis been performed appropriately and rigorously? 

Reviewer #1: N/A

4. Have the authors made all data underlying the findings in their manuscript fully available?

Reviewer #1: Yes

5. Is the manuscript presented in an intelligible fashion and written in standard English?

Reviewer #1: Yes

6. Review Comments to the Author

Reviewer #1: Thank you for your thoughtful and comprehensive revisions. i believe your research makes an important contribution to the literature.

There is one revision that needs further refining:

Although the findings on the effects of male partners’ presence during childbirth remain

equivocal, some studies have reported that the involvement of male partners during childbirth

contributes to a positive maternal outcomes such as emotional stability, a relieved aftermath

depression, a realized birth expectations, and a faster recovery through the enhanced adherence to

medical care and provision of nutritional support to the woman immediately following childbirth.

(12–14).

Please find alternative phrasing to explain 'relieved aftermath depression' (eg do you mean associated with lower levels of depression in the postpartum period? improved recovery?) and 'a realized birth expectations'.

7. PLOS authors have the option to publish the peer review history of their article (what does this mean?). If published, this will include your full peer review and any attached files.

Reviewer #1: No

---

## [Author Response · Author response to Decision Letter 1]

3 May 2023

Patrick G Bagambe

Senior Lecturer of Obstetrics and Gynecology

University of Rwanda

patrickgatsinzi.pg@gmail.com

+250 788 302 804

August 23, 2022/ New response (26/04/2023).

Dear Sir/Madam

Re: PLOS-ONE-D-21-39209: “Perceptions of male partners on maternal near-miss events experienced by their female partners in Rwanda”. 

Please find enclosed a revised version of our manuscript number PLOS-ONE-D-21-39209. The Revisions are based comments and suggestions from yourself and the reviewers. The manuscript has been revised considerably. Please find enclosed a detailed response to the comments the manuscript reviewers.

The manuscript has not been submitted for consideration for publication elsewhere. There are no financial agreements or other conflicts of interest to disclose.

I look forward to your evaluation of the revised manuscript. 

Sincerely,

Patrick Bagambe

Response to Reviewers and editors' comments:

Interview guide.

The interview guide should be described in the methods section briefly.

The interview guide has been described in the methods section

A description of the coding tree with theme(s) should be provided.

The Coding tree has been described in the methodology section and the six emerging themes are documented in the results section

The authors should state how many meaning units and codes were extracted from the data.

64 meaning units were extracted and condensed into 25 codes. Those codes make the six broad themes.

In page 4, the authors state, ‘Once a woman with near-miss was identified from the TSAM registry, we checked the hospital registry to identify her local village and we used the Community Health Worker’s …….’ But in the limitations of the study we find that the interviews were conducted months after discharging the women from the hospital. For the sake of clarity, the interval between the discharge of a mother from hospital and the interview with her spouse should be included in table 1 or 2.

The time interval from maternal discharge to the interview with the male partner was 12 to 24 months among 12 participants but some were interviewed after two years. More details were summarized in table 2.

Please find alternative phrasing to explain 'relieved aftermath depression' (eg do you mean associated with lower levels of depression in the postpartum period? improved recovery?) and 'a realized birth expectations'.

Alternative phrasing has been added

---

## [Decision Letter · Decision Letter 2]

23 May 2023

Perceptions of male partners on maternal near-miss events experienced by their female partners in Rwanda

PONE-D-21-39209R2

Dear Dr. Bagambe,

We’re pleased to inform you that your manuscript has been judged scientifically suitable for publication and will be formally accepted for publication once it meets all outstanding technical requirements.

Kind regards,

Forough Mortazavi

Academic Editor

PLOS ONE

Additional Editor Comments (optional):

Reviewers' comments:

Reviewer's Responses to Questions

**Comments to the Author**

1. If the authors have adequately addressed your comments raised in a previous round of review and you feel that this manuscript is now acceptable for publication, you may indicate that here to bypass the “Comments to the Author” section, enter your conflict of interest statement in the “Confidential to Editor” section, and submit your "Accept" recommendation.

Reviewer #1: (No Response)

2. Is the manuscript technically sound, and do the data support the conclusions?

Reviewer #1: Yes

3. Has the statistical analysis been performed appropriately and rigorously? 

Reviewer #1: N/A

4. Have the authors made all data underlying the findings in their manuscript fully available?

Reviewer #1: Yes

5. Is the manuscript presented in an intelligible fashion and written in standard English?

Reviewer #1: Yes

6. Review Comments to the Author

Reviewer #1: Thank you for conducting this research and offering a platform for theses voices to be heard. I am satisfied with the revisions

7. PLOS authors have the option to publish the peer review history of their article (what does this mean?). If published, this will include your full peer review and any attached files.

Reviewer #1: No

---

## [Editor Report · Acceptance letter]

31 May 2023

PONE-D-21-39209R2 

Perceptions of male partners on maternal near-miss events experienced by their female partners in Rwanda 

Dear Dr. Bagambe:

I'm pleased to inform you that your manuscript has been deemed suitable for publication in PLOS ONE. Congratulations! Your manuscript is now with our production department. 

Kind regards, 

on behalf of

Dr. Forough Mortazavi 

Academic Editor

PLOS ONE